# Unhealthy food consumption among 20–59 years old adults in Bangladesh: Findings from a nationally representative cross-sectional survey

Shahnaz Sharmin [1], Fahmida Akter [1], Md. Mokbul Hossain [1], Abu Ahmed Shamim[1], Abu Abdullah Mohammad Hanif[1], Mehedi Hasan[1], Md Showkat Ali Khan[1], Mohammad Aman Ullah[2], Dipak Kumar Mitra [3], Malay Kanti Mridha [1]*

1 Centre for Non-communicable Diseases and Nutrition, BRAC James P Grant School of Public Health, BRAC University, Dhaka, Bangladesh, 2 National Nutrition Services (NNS), Institute of Public Health Nutrition (IPHN), Dhaka, Bangladesh, 3 North-South University, Dhaka, Bangladesh

* malay.mridha@bracu.ac.bd

## Abstract

### Background

Unhealthy food consumption is a major public health concern because of its adverse health consequences. We assessed unhealthy food (savory and fried snacks, SFS; sweet snacks, SS; and sugar-sweetened beverages, SSBs) consumption patterns and identified factors associated with this consumption among adults aged 20–59 years in Bangladesh.

### Methods

Data from the Bangladesh Food Security and Nutrition Surveillance Project (2018–2019) were used in this study. We interviewed 4,917 men and 5,069 women (aged 20–59 years) from 82 sentinel sites, selected through multistage cluster sampling from eight administrative divisions. A structured questionnaire was used to collect data. Descriptive analysis and Poisson regression with robust variance were performed to report prevalence and adjusted prevalence ratio (APR).

### Results

The weighted prevalence of weekly SFS, SS, and SSBs consumption was 57.8%, 75.2%, and 77.3% among men and 29.6%, 58.7%, and 35.3% among women, respectively. SFS, SS, and SSBs consumption were associated with education and wealth quintiles in both men and women. Being a current smoker was associated with SFS, SS, and SSBs consumption only among men. Marital status, inadequate fruit and vegetable intake, overweight/obesity, high sedentary time, hypertension, and self-reported asthma were associated with SSBs consumption only among women. Age was associated with SFS consumption in both men and women. SFS and SSBs

**Data availability statement:** All relevant data are within the paper and its Supporting Information files.

**Funding:** MKM received the fund. This study was funded by the National Nutrition Services (NNS), Institute of Public Health Nutrition, Ministry of Health and Family Welfare, Government of Bangladesh (Memo: 45.165.032.01.00.003.2016-325; Date: 10 December 2017). URL: https://www.nutrition-map.org/. The funders were involved with the conceptualization of the study design, data supervision, and reviewing the manuscript.

**Competing interests:** The authors have declared that no competing interests exist.

consumption was also associated with place of residence in both men and women. Inadequate fruit and vegetable intake was associated with a lower prevalence ratio with SFS consumption in both men and women, and diabetes was associated with a lower prevalence ratio with SSBs intake in men.

## Conclusions

We found a high prevalence of SFS, SS, and SSBs consumption among 20–59 years old adults in Bangladesh. Several sociodemographic, behavioral, and clinical factors were associated with unhealthy food consumption. These factors should be considered while designing and implementing interventions to reduce unhealthy food consumption in adults.

## Introduction

Globally, consumption of unhealthy food and sugar-sweetened beverages (SSBs) is rising [1] and has become a major public health concern because of their adverse impact on health [2,3]. A rapid transition from healthy food consumption to unhealthy food such as junk food or fast-food consumption is happening around the word [4]. The factors behind this transition include urbanization, widespread marketing by the food industry, and economic growth [4]. The consumption of unhealthy food is considered a major driver of overweight or obesity because of its inappropriate nutritional content [5], and these unhealthy foods are often associated with NCDs [6].

According to the World Health Organization (WHO), the recommended free sugar intake for adults as well as children should be less than 10% of the total energy intake, and 5% per day will be good for health [7]. However, according to research, in 2010, the global burden of diseases related to SSBs intake was 8.5 million disability adjusted life years, attributable to 184,000 deaths [8]. In 2010, mean global SSBs intake was 0.58 servings/day, and in high-income countries (HICs), upper-middle, lower-middle, and low-income countries (LICs) the intake was 0.51, 0.80, 0.59 and 0.35 servings/day, respectively [8]. SSB intake has risen in both HICs and low- and middle-income countries (LMICs) [9]. Nevertheless, the alarming fact is the growing rate of unhealthy food intake in LMICs is even faster than that in HICs [10]. In South Asian countries, an increasing trend is evident, and in Bangladesh, the consumption of junk food is constantly increasing [11]. In addition, the consumption of SFS, namely chips, *chanachur*, fried pulses, *peyaju*, and *singara*, as well as SS, such as biscuits, cakes, and ice cream, is common in Bangladesh [12]. These foods typically have high levels of harmful ingredients, such as salt, sugar, and saturated fat, but lack health-promoting ingredients such as dietary fiber [13].

Several factors are associated with the consumption of unhealthy foods such as fast food or junk food. Educational attainment, time constraints, new taste and routine [14], better taste, easy accessibility, and low price are factors associated with fast food or junk food consumption [15]. On the other hand, factors related to SSBs intake include sex, age, wealth index, marital status [16], lower socioeconomic status, low

level of education, unemployment, smoking, physical inactivity, poor dietary habits [17], availability, affordability, advertising, electronic device use, and social gatherings [18]. Identification of these associated factors are needed, as they influence the intake of unhealthy food and SSBs, which may lead to further health complications.

Existing evidence indicates that an unhealthy diet is a key determinant of rising obesity, diabetes, hypertension, and cardiovascular disease (CVD) [19]. It has already been established that SSBs intake contributes to obesity [3], which is related to type 2 diabetes, cancer, kidney, heart, and liver disease, dental caries, and other medical and mental health diagnoses [3,20,21]. In Bangladesh, 73.2% of all deaths occurred due to NCDs in 2017 [22].

To date, the majority of research conducted in Bangladesh on unhealthy food and beverage intake, was focused on children and adolescents [15,23,24]. There are no or very limited nationally representative data on the prevalence of consumption of SFS, SS, and SSBs among men and women aged 20–59 years, although this life stage is the most productive and spans between adolescence and senescence. Moreover, there are hardly any findings regarding the factors associated with the consumption of these food items among these population groups in Bangladesh. Accordingly, given the harmful consequences of unhealthy food intake on health, it is essential to determine the prevalence of consumption and factors associated with SFS, SS, and SSBs. Therefore, this study was designed to assess the level of unhealthy food consumption and identify the factors associated with the consumption of these foods among women and men aged 20–59 years in Bangladesh. The findings of this study will help policymakers by providing evidence to develop strategies to reduce the consumption of unhealthy foods and SSBs as well as related morbidities and mortalities.

## Methods

### Study design and site

This study utilized Food Security and Nutrition Surveillance Project (FSNSP) 2018–2019 data from Bangladesh, which was a cross-sectional survey and carried out from October 2018 to October 2019. The study design has been described in several papers [25,26]. In summary, this study was conducted to estimate different nutritional and health-related variables among six population categories, involving the 20–59 years old adult men and women. Participants in this study were selected from eight administrative divisions and all types of residences in Bangladesh, including rural, non-slum urban, and slum areas.

### Sample size and sampling techniques

The sample size was determined to obtain representative estimates at the national and divisional levels for the required variables, with a prevalence ranging from 4% to 98%. We estimated 62 adult participants as the required sample size for each cluster, resulting in a total of 5580 adults from a random selection of 90 clusters (64 rural, 16 urban, and 10 slum). Multistage cluster sampling techniques were applied to select study participants in rural, non-slum urban, and slum clusters. Further details of the sampling technique and sample size have been reported elsewhere [26]. **Fig 1** details the selection of study subjects for this study.

### Data collection tool and procedure

The data were collected using structured questionnaires. To measure the weight, height, and blood pressure of the study subjects, a portable weighing scale (TANITA, UM-070), a locally made height board, and a blood pressure machine (Omron, HEM 7120) were used. Data regarding the consumption of savory and fried snacks, sweets, and SSBs, behavioral factors, self-reported NCDs, other anthropometric measurements (height, weight), and blood pressure were recorded in the participant's residence. Prior to data collection, the data collectors were trained to conduct interviews and perform anthropometric measurements accurately.

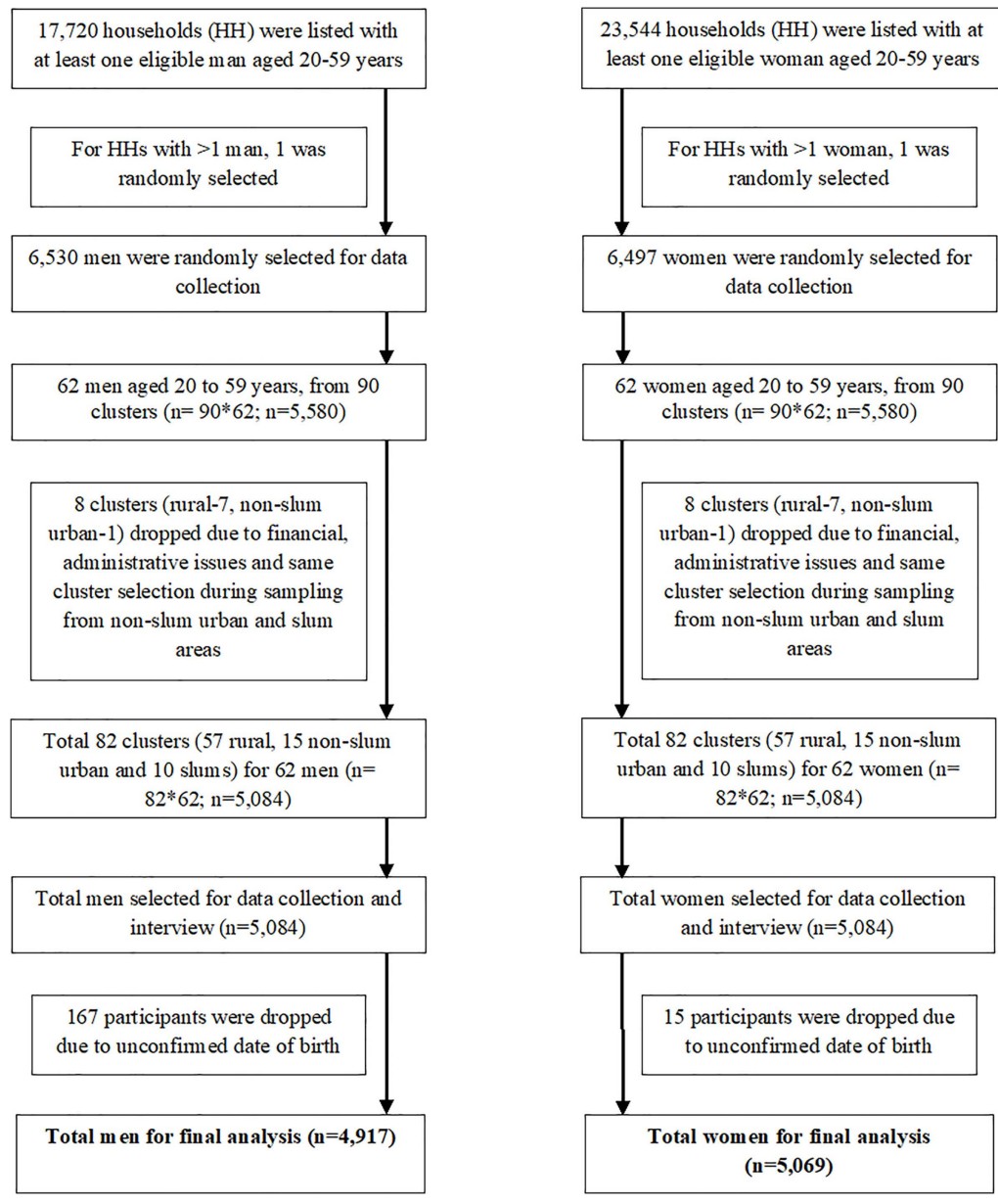

**Fig 1. Flowchart displaying selection of study participants.**

## Quality control

To assure quality, the data were reviewed and cross-checked. During the survey, field coordinators reviewed the question-naires to identify any errors and immediately corrected them in the field. The statistician generated data queries daily. More-over, the calibration of the blood pressure machine and other anthropometric instruments was performed on a regular basis.

## Study variable

**Outcome variable.** The questionnaire included three categories of unhealthy foods (savory and fried snacks, sweets, and SSB) that were listed in the data collection tools for measuring the dietary diversity of women [27]. SFS consisted

of spicy or salty snacks. These snacks included commonly consumed deep-fried snacks, such as pakoras, samosas, *singara*, chips, *chanachur,* and fried pulses. A detailed description of these foods can be found in a different publication [12]. SS included a variety of traditional milk-based confections from South Asia, as well as homemade desserts such as halwa, which is made by boiling semolina, water, or milk, adding sugar, oil, or butter, and aromatic spices such as cardamom and cinnamon. Additionally, SS foods included snacks like biscuits, cakes, chocolate, and sweets or candy from restaurants or grocery stores [28]. SSBs are described as any beverage with added sugar or include soft drinks, energy drinks, sports drinks, or any kind of sweetened milk, tea or coffee [29].

The main outcome measure was the consumption of the SFS, SS, and SSBs. Consumption was measured by asking, during the last 7 days, how many days they consumed SFS, SS, and SSBs. The responses were classified into two categories '0' never and '1' weekly consumption (at least one day in the last 7 days preceding the interview). To calculate the prevalence and identify the factors related to SFS, SS, and SSBs, the above-mentioned categories were used.

**Explanatory variables.** Explanatory variables were chosen based on the existing literature and data availability of the study. The operational definitions of the explanatory variables, including sociodemographic, behavioral, and clinical variables, are given in (Table 1).

Table 1. List of explanatory variables for analysis.

| Variables | Description |
|---|---|
| Age | 20-29, 30–39, 40–49, 50–59 years |
| Place of residence | Rural, non-slum urban, slum |
| Division | Dhaka, Chattogram, Rajshahi, Khulna, Barisal, Sylhet, Rangpur, Mymensingh |
| Educational status | No formal education, primary, secondary, higher secondary & above |
| Occupation | Not working/homemaker, working |
| Wealth index | Poorest, poorer, middle, richer, richest |
| Marital status | Currently married, others (never married, separated, divorced, widowed) |
| Religion | Islam, others (Hindu, Christian, Buddhist) |
| Physical activity | Sufficient (≥150 minutes of moderate-intensity or ≥75 minutes of vigorous-intensity physical activity or an equivalent combination of both in a week) Insufficient (<150 minutes of moderate-intensity or <75 minutes of vigorous-intensity physical activity or an equivalent combination of both in a week) |
| Sedentary time | <=7 hours,> 7 hours |
| Duration of watching television | <=4 hours,> 4 hours |
| Fruits and vegetable consumption | Adequate (≥5 servings per day), inadequate (<5 servings per day) |
| Currently smoking | Yes, no |
| BMI | Underweight (<18.5 kg/m2), normal (≥18.5 to <23.0 kg/m2), overweight (≥23.0 to <27.5 kg/m2) and obese (≥27.5 kg/m2) |
| Hypertension | Hypertensive (SBP ≥ 140 mm Hg or DBP ≥ 90 or the participants were diagnosed with hypertension by a medical professional), non-hypertensive (SBP < 140mm Hg or DBP < 90 or the participants were not diagnosed with hypertension by a medical professional) |
| Self-reported diabetes | Yes, no |
| Self-reported heart diseases | Yes, no |
| Self-reported asthma | Yes, no |

## Data analysis

Data management and analysis were carried out using Stata V.17.0 (StataCorp). Descriptive analysis was performed to estimate the level of consumption based on sociodemographic, behavioral, and clinical factors. The weighted prevalence of savory and fried snacks, SS and SSBs consumption was estimated using 95% confidence interval (CI). Poisson regression, with robust variance was carried out to estimate the unadjusted and adjusted prevalence ratio (PR) of SFS, SS and SSBs consumption; as the prevalence of consumption for all the food types were >10% [30]. In cross-sectional studies, the odds ratio (OR) may overestimate the magnitude of association in comparison with PR [31] when the prevalence is high. Moreover, OR may have a wider CI, which leads to less precise estimation [31].Variables with a p value ≤ 0.2 in the unadjusted analysis were included in the adjusted analysis [32]. Model fitness was assessed using Pearson's chi-square and deviance goodness-of-fit tests. The variance inflation factor was checked to assess the multicollinearity among the explanatory variables. In addition, Bonferroni correction was used. Statistical significance was considered when the p-value was < 0.05.

## Ethical consideration

Ethical approval from the Institutional Review Board (IRB) of the BRAC James P Grant School of Public Health, BRAC University, Dhaka, Bangladesh (IRB reference number: 2018–020-IR) was obtained for the study. Prior to data collection, participants were informed about the study objective, and it was also assured that they had the right to refuse or discontinue the interview at any moment. Written consent was obtained from all participants.

## Results

### Characteristics of the study participants

A total of 9,986 participants were included in the analysis, of which 4,917 (49.2%) were men and 5,069 (50.8%) were women. One-third of the participants (33.4%) were aged 30–39 years age category, and more than 70% were from rural areas. About 86% of the participants belonged to Islam, and 89% were married. Thirty percent of the respondents had no formal education and more than half (51.3%) were working. Approximately 19% of the participants reported less than 150 min of physical activity per week, of which 23.9% were men and 13.7% were women. Overall, 24.3% of the respondents spent more than seven hours of sedentary time per day (men: 26.4%; women: 22.2%). The majority (83.0%) of the respondents did not consume adequate fruits and vegetables, while 77.2% were men and 88.6% were women. Approximately one-fourth of the study participants (24.4%) were current smokers. Forty-five percent of the respondents were overweight or obese (men, 36.1%; women, 53.6%). About 22% of the participants had hypertension, whereas the proportions of self-reported heart disease, asthma, and diabetes were 7.5%, 5.9%, and 4.9%, respectively (**Table 2**).

Prevalence of SFS, SS and SSBs consumption in the last week

The prevalence of SFS consumption among men was 57.8%, which was nearly twice (29.6%) compared to women (Fig 2). In both men and women, the prevalence of SFS consumption was higher in the 20–29 years age category (men 69.3% and women 35.4%) than in those belonging to other age categories. Among the divisions, respondents in Chattogram were more likely to consume SFS (men 82.3% and women 49.7%) than respondents in Rajshahi (men 42.3% and women 16.6%). The level of SFS consumption was highest among non-slum urban men (69.0%) and slum-dwelling women (48.4%). The respondents with higher secondary and higher education (67.6% men and 46.1% women) and those belonging to the richest wealth quintile (62.7% men and 43.5% women) had higher consumption of SFS. SFS consumption was also more prevalent among men and women who watched television for more than four hours a day and spent more than seven hours of sedentary time (S1Table).

The prevalence rates of SS consumption among men and women were 75.2% and 58.7%, respectively. The prevalence of SS consumption declined with age in both sexes. A higher prevalence of SS consumption was found in the Sylhet

**Table 2. Background characteristics of the study participants.**

| Variables | Total (n = 9,986) | | Men (n = 4,917) | | Women (n = 5,069) | |
|---|---|---|---|---|---|---|
| | n | % | n | % | n | % |
| **Age in years** | | | | | | |
| 20-29 | 2835 | 28.4 | 1243 | 25.3 | 1592 | 31.4 |
| 30-39 | 3332 | 33.4 | 1572 | 32.0 | 1760 | 34.7 |
| 40-49 | 2327 | 23.3 | 1243 | 25.3 | 1084 | 21.4 |
| 50-59 | 1492 | 14.9 | 859 | 17.5 | 633 | 12.5 |
| **Division** | | | | | | |
| Dhaka | 1232 | 12.3 | 567 | 11.5 | 665 | 13.1 |
| Chattogram | 1483 | 14.9 | 743 | 15.1 | 740 | 14.6 |
| Rajshahi | 1357 | 13.6 | 677 | 13.8 | 680 | 13.4 |
| Khulna | 1362 | 13.6 | 682 | 13.9 | 680 | 13.4 |
| Barisal | 977 | 9.8 | 483 | 9.8 | 494 | 9.7 |
| Sylhet | 854 | 8.6 | 420 | 8.5 | 434 | 8.6 |
| Rangpur | 1358 | 13.6 | 674 | 13.7 | 684 | 13.5 |
| Mymensingh | 1363 | 13.6 | 671 | 13.6 | 692 | 13.7 |
| **Place of residence** | | | | | | |
| Rural | 7017 | 70.3 | 3483 | 70.8 | 3534 | 69.7 |
| Non-slum urban | 1744 | 17.5 | 833 | 16.9 | 911 | 18 |
| Slum | 1225 | 12.3 | 601 | 12.2 | 624 | 12.3 |
| **Religion** | | | | | | |
| Islam | 8565 | 85.8 | 4195 | 85.3 | 4370 | 86.2 |
| Others[a] | 1419 | 14.2 | 721 | 14.7 | 698 | 13.8 |
| **Marital status** | | | | | | |
| Currently married | 8862 | 88.7 | 4247 | 86.4 | 4615 | 91.0 |
| Others[b] | 1124 | 11.3 | 670 | 13.6 | 454 | 9.0 |
| **Education** | | | | | | |
| No formal education | 2976 | 29.8 | 1424 | 29.0 | 1552 | 30.6 |
| Primary (grade 1–5) | 2835 | 28.4 | 1404 | 28.6 | 1431 | 28.2 |
| Secondary (grade 6–10) | 2961 | 29.7 | 1328 | 27.0 | 1633 | 32.2 |
| Higher secondary & above | 1214 | 12.2 | 761 | 15.5 | 453 | 8.9 |
| **Occupation** | | | | | | |
| Not working/homemaker | 4865 | 48.7 | 347 | 7.1 | 4518 | 89.1 |
| Working | 5121 | 51.3 | 4570 | 92.9 | 551 | 10.9 |
| **Wealth quintile** | | | | | | |
| Poorest | 2048 | 20.5 | 987 | 20.1 | 1061 | 20.9 |
| Poorer | 1995 | 20.0 | 982 | 20.0 | 1013 | 20.0 |
| Middle | 1942 | 19.5 | 985 | 20.0 | 957 | 18.9 |
| Richer | 2023 | 20.3 | 983 | 20.0 | 1040 | 20.5 |
| Richest | 1975 | 19.8 | 979 | 19.9 | 996 | 19.7 |
| **Physical activity** | | | | | | |
| >=150 Minutes/week | 8118 | 81.3 | 3744 | 76.1 | 4374 | 86.3 |
| < 150 Minutes/week | 1868 | 18.7 | 1173 | 23.9 | 695 | 13.7 |
| **Fruits and vegetables intake** | | | | | | |
| >= 5 ser/day | 1701 | 17.0 | 1122 | 22.8 | 579 | 11.4 |
| <5 ser/day | 8285 | 83.0 | 3795 | 77.2 | 4490 | 88.6 |

*(Continued)*

**Table 2.** (Continued)

| Variables | Total (n = 9,986) | | Men (n = 4,917) | | Women (n = 5,069) | |
|---|---|---|---|---|---|---|
| | n | % | n | % | n | % |
| **Sedentary time** | | | | | | |
| <=7 hours/day | 7562 | 75.7 | 3620 | 73.6 | 3942 | 77.8 |
| > 7hours/day | 2424 | 24.3 | 1297 | 26.4 | 1127 | 22.2 |
| **Duration of watching TV** | | | | | | |
| <=4 hours/day | 9203 | 92.2 | 4543 | 92.4 | 4660 | 91.9 |
| > 4 hours/day | 783 | 7.8 | 374 | 7.6 | 409 | 8.1 |
| **Current smoking** | | | | | | |
| No | 7551 | 75.6 | 2526 | 51.4 | 5025 | 99.1 |
| Yes | 2435 | 24.4 | 2391 | 48.6 | 44 | 0.9 |
| **Body mass index (BMI)** | | | | | | |
| Underweight | 1267 | 12.7 | 757 | 15.4 | 510 | 10.1 |
| Normal | 4221 | 42.3 | 2383 | 48.5 | 1838 | 36.3 |
| Overweight and/or obese | 4484 | 45.0 | 1772 | 36.1 | 2712 | 53.6 |
| **Hypertension** | | | | | | |
| Non-hypertensive | 7824 | 78.3 | 3975 | 80.8 | 3849 | 75.9 |
| Hypertensive | 2162 | 21.7 | 942 | 19.2 | 1220 | 24.1 |
| **Self -reported heart disease** | | | | | | |
| No | 9237 | 92.5 | 4596 | 93.5 | 4641 | 91.6 |
| Yes | 749 | 7.5 | 321 | 6.5 | 428 | 8.4 |
| **Self- reported asthma** | | | | | | |
| No | 9396 | 94.1 | 4660 | 94.8 | 4736 | 93.4 |
| Yes | 590 | 5.9 | 257 | 5.2 | 333 | 6.6 |
| **Self- reported diabetes** | | | | | | |
| No | 9496 | 95.1 | 4727 | 96.1 | 4769 | 94.1 |
| Yes | 490 | 4.9 | 190 | 3.9 | 300 | 5.9 |

[a]Hindu, Christian, Buddhist together

[b]Never married, separated, divorced, widowed

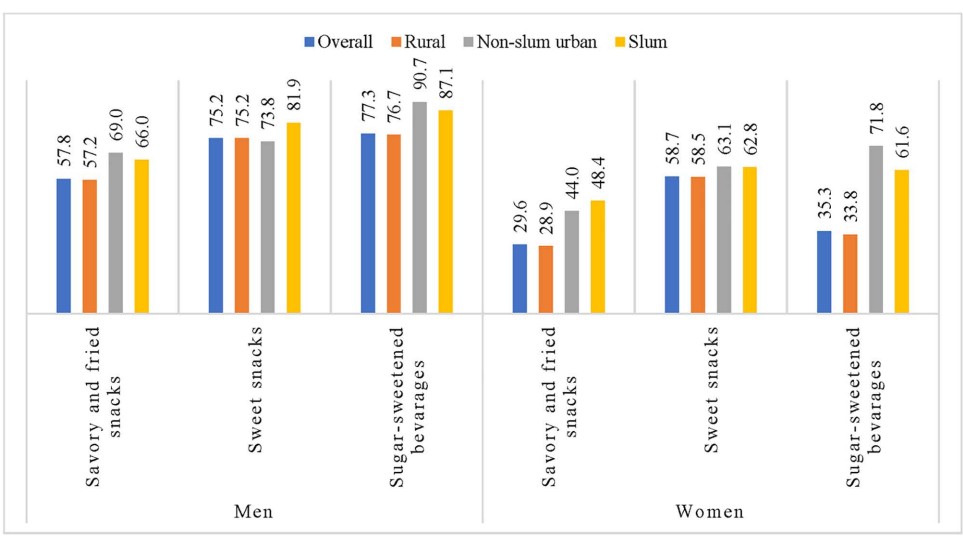

**Fig 2. Prevalence of SFS, SS, and SSBs consumption (last 7 days) by gender and area of residence.**

division (89.8%), whereas for women, it was in the Chattogram division (83.2%). No major variation in the prevalence of SS consumption was found according to place of residence, religion, marital status, or occupation among women. However, men living in slums (82.0%) had a higher prevalence compared to those living in non-slum urban areas (73.8%). Additionally, men who were unmarried, divorced, separated, or widowed (83.2%) had a higher prevalence of SS consumption than those who were married (73.5%). Both men and women with higher education and the richest wealth quintile reported more consumption of SS than their counterparts (S1 Table).

The prevalence of SSBs intake among men was more than twofold (77.3%) compared to women (35.3%) (Fig 2). Both men and women living in the Chattogram division (men, 95.2%; women, 89.0%) were more likely to report SSBs consumption than those in the other divisions. The consumption prevalence for both sexes was found to be higher in non-slum urban areas (men, 90.7%; women, 71.8%) than in their counterparts. The consumption of SSBs among men and women increased with education. Men in the poorer wealth quantile (82.5%) and women in the richest wealth quintile (53.1%) reported higher SSBs consumption. SSBs consumption was found to be higher among women who performed less than 150 minutes per week of physical activity, although the consumption remained almost the same for men who did or did not perform 150 minutes per week of physical activity. Respondents who were smokers had a higher prevalence of SSB consumption (men, 87.8%; women, 62.8%) than those who were non-smokers (men, 70.0%; women, 35.2%) (S1 Table).

## Multicollinearity and goodness-of-fit statistics

Multicollinearity statistics for explanatory variables and goodness-of-fit statistics for SFS, SS, and SSBs for adult men and women are presented in Tables S2, S3, and S4, respectively. The mean VIF values were less than 10 for all models considered, indicating no multicollinearity among the explanatory variables. Both deviance and Pearson Chi-square statistics showed large p-values, indicating that there was no lack of fit, suggesting that the Poisson regression models fit the data well.

## Factors associated with SFS consumption

The crude and adjusted prevalence ratios for women and men are presented in Figure 3. From the adjusted analysis among the men, age group 20–29 years old (APR: 1.39, 95% CI: 1.28, 1.52, p < 0.001), 30–39 years old (APR: 1.23, 95% CI: 1.13, 1.33, p < 0.001) and 40–49 years old (APR: 1.15, 95% CI: 1.06, 1.25, p < 0.01); staying in Chattogram division (APR: 1.12, 95% CI: 1.05, 1.20, p < 0.01); residing in non-slum urban area (APR: 1.11, 95% CI: 1.03, 1.19, p < 0.01) and slum area (APR: 1.12, 95% CI: 1.05, 1.20, p < 0.01); having primary education (APR: 1.14, 95% CI: 1.07, 1.22, p < 0.001) secondary education (APR: 1.22, 95% CI: 1.14, 1.30, p < 0.001) and higher secondary and above education (APR: 1.28, 95% CI: 1.17, 1.39, p < 0.001); belong to poorer wealth quintile (APR: 1.12, 95% CI: 1.04, 1.20, p < 0.01), middle (APR: 1.11, 95% CI: 1.04, 1.20, p < 0.01), richer (APR: 1.08, 95% CI: 1.00, 1.17, p < 0.05) and richest (APR: 1.10, 95% CI: 1.02, 1.19, p < 0.05); being current smoker (APR: 1.10, 95% CI: 1.05, 1.15, p < 0.001) were associated with higher prevalence of SFS consumption (Fig 3).

From the adjusted analysis among the women, age group 20–29 years old (APR: 1.36, 95% CI: 1.15, 1.61, p < 0.001), 30–39 years old (APR: 1.25, 95% CI: 1.07, 1.47, p < 0.01); staying in Chattogram division (APR: 1.16, 95% CI: 1.02, 1.31, p < 0.05) and Barisal (APR: 1.27, 95% CI: 1.11, 1.46, p < 0.01); residing in non-slum urban area (APR: 1.19, 95% CI: 1.06, 1.35, p < 0.01) and slum (APR: 1.18, 95% CI: 1.05, 1.33, p < 0.01); having primary education (APR: 1.20, 95% CI: 1.07, 1.35, p < 0.01) secondary education (APR: 1.38, 95% CI: 1.23, 1.56, p < 0.001) and higher secondary and above education (APR: 1.63, 95% CI: 1.41, 1.89, p < 0.001); belong to poorer wealth quintile (APR: 1.15, 95% CI: 1.01, 1.32, p < 0.05), middle (APR: 1.15, 95% CI: 1.01, 1.32, p < 0.05), richer (APR: 1.19, 95% CI: 1.04, 1.36, p < 0.05) and richest (APR: 1.34, 95% CI: 1.16, 1.53, p < 0.001) were associated with higher prevalence of SFS consumption. Insufficient (<5 servings per day) fruit and vegetable intake (APR: 0.81, 95% CI: 0.72, 0.91, p < 0.001) was associated with a lower prevalence of SFS consumption (Fig 3).

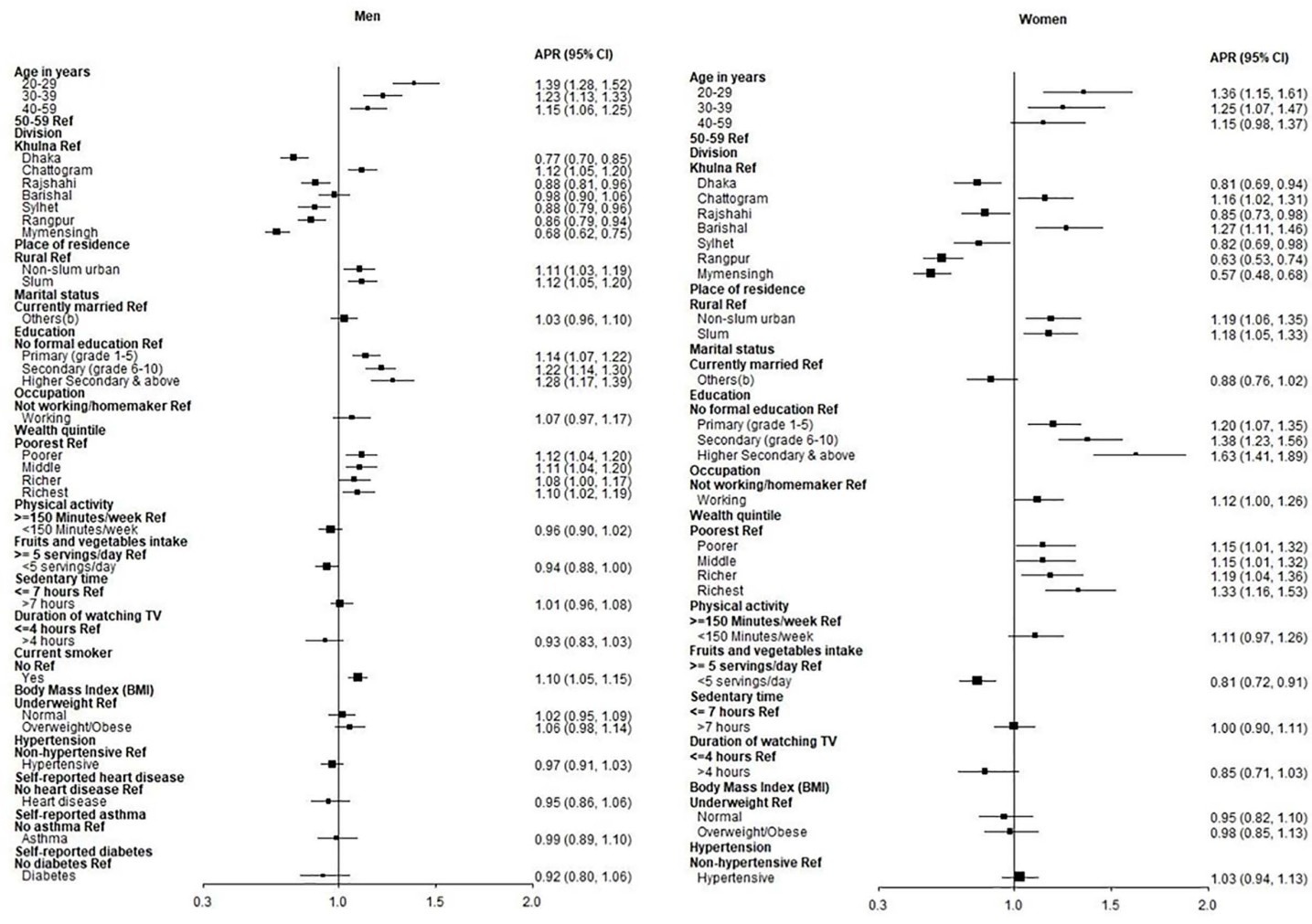

**Fig 3. Forest plot of adjusted prevalence ratios (APR) for factors of SFS consumption among men and women.**

Bonferroni corrected p values and confidence interval (CI) for SFS among men and women is provided in S8 Table.

## Factors associated with SS consumption

From the adjusted analysis among the men, age group 20–29 years old (APR: 1.06, 95% CI: 1.01, 1.12, p < 0.05); having primary education (APR: 1.07, 95% CI: 1.02, 1.11, p < 0.01), secondary education (APR: 1.10, 95% CI: 1.06, 1.15, p < 0.00), and higher secondary and higher education (APR: 1.18, 95% CI: 1.12, 1.24, p < 0.001); belonging to richer wealth quintile (APR: 1.05, 95% CI: 1.00, 1.10, p < 0.05) and richest (APR: 1.08, 95% CI: 1.03, 1.13, p < 0.01); and being a current smoker (APR: 1.05, 95% CI: 1.02, 1.08, p < 0.01) were associated with a higher prevalence of SS. However, insufficient physical activity (APR: 0.92, 95% CI: 0.89, 0.96, p < 0.001) and watching television for more than 4 h/day (APR: 0.92, 95% CI: 0.84, 1.00, p < 0.05) were associated with a lower prevalence of SS consumption (**Fig 4**).

From the adjusted analysis among the women staying in Sylhet division (APR: 1.18, 95% CI: 1.10, 1.27, p < 0.001); having primary education (APR: 1.07, 95% CI: 1.01, 1.14, p < 0.05) secondary education (APR: 1.20, 95% CI: 1.13, 1.28, p < 0.001) and higher secondary and above education (APR: 1.39, 95% CI: 1.29, 1.50, p < 0.001); belong to middle wealth

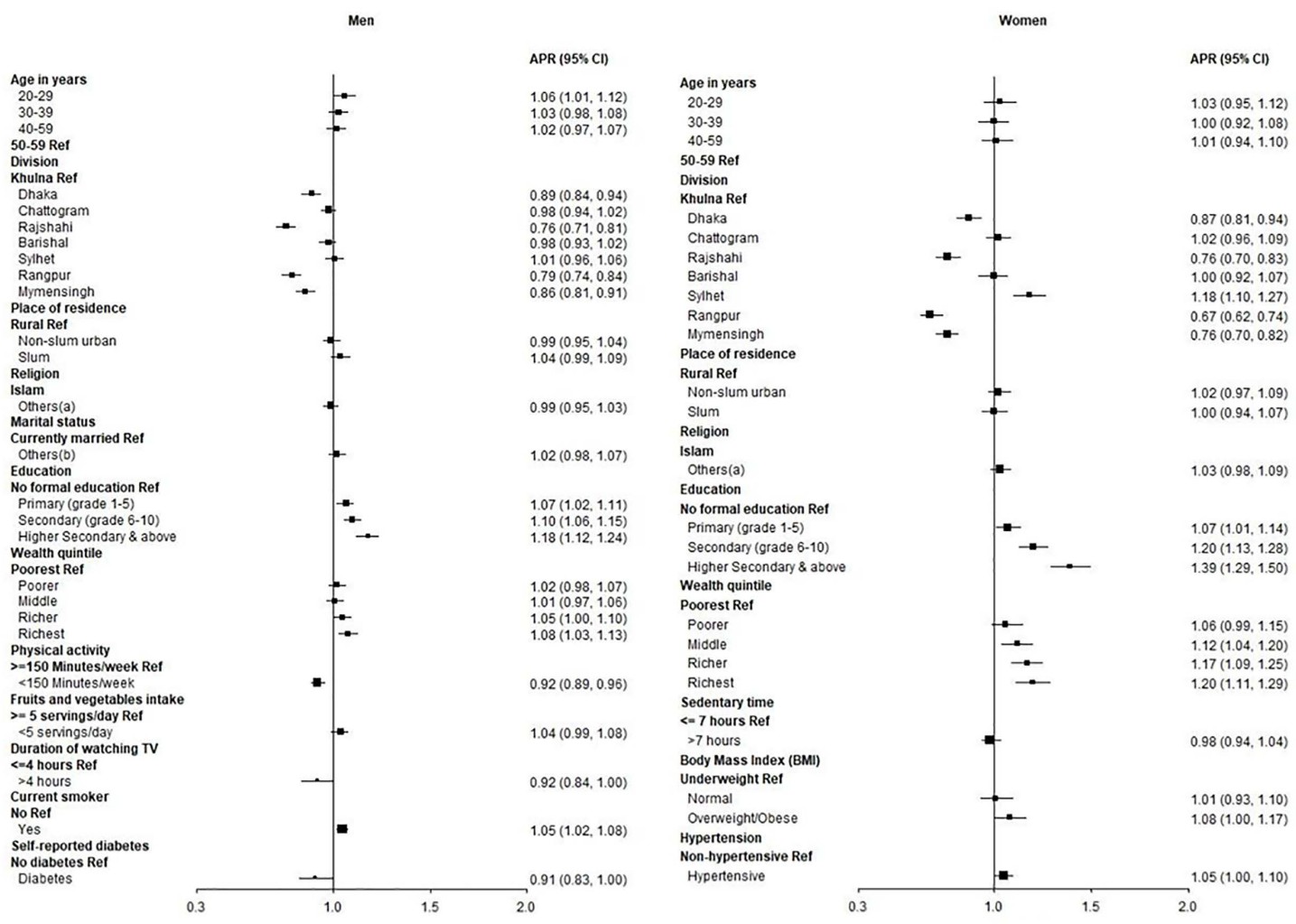

**Fig 4. Forest plot of adjusted prevalence ratios (APR) for factors of SS consumption among men and women.**

quintile (APR: 1.12, 95% CI: 1.04, 1.20, p<0.01), richer (APR: 1.17, 95% CI: 1.09, 1.25, p<0.001) and richest (APR: 1.20, 95% CI: 1.11, 1.29, p<0.001); and being overweight/obese (APR: 1.08, 95% CI: 1.00, 1.17, p<0.05) were associated with higher prevalence of SS consumption (**Fig 4**).

Bonferroni corrected p values and confidence interval (CI) for SS among adult men and women is provided in S9 Table.

## Factors associated with SSBs consumption

From the adjusted analysis among the men, living in Dhaka division (APR: 1.06, 95% CI: 1.00, 1.13, p<0.05), Chattogram (APR: 1.21, 95% CI: 1.15, 1.26, p<0.001), Barisal (APR: 1.22, 95% CI: 1.16, 1.29, p<0.001), Sylhet (APR: 1.22, 95% CI: 1.16, 1.29, p<0.001), Rangpur (APR: 1.16, 95% CI: 1.10, 1.22, p<0.001) and Mymensingh (APR: 1.13, 95% CI: 1.07, 1.19, p<0.001); living in non-slum urban area (APR: 1.12, 95% CI: 1.08, 1.16, p<0.001) and slum (APR: 1.08, 95% CI: 1.05, 1.12, p<0.001); having secondary education (APR: 1.04, 95% CI: 1.01, 1.08, p<0.05), higher secondary and above education (APR: 1.11, 95% CI: 1.06, 1.15, p<0.001); belong to the richest wealth quintile (APR: 1.04, 95% CI: 1.00, 1.09, p<0.05); and being current smoker (APR: 1.16, 95% CI: 1.13, 1.18, p<0.001) were associated with higher prevalence of

SSBs consumption. Self-reported diabetes (APR: 0.80, 95% CI: 0.73, 0.88, p<0.001) was associated with a lower prevalence of SSBs consumption (Fig 5).

From the adjusted analysis among the women, living in Dhaka division (APR: 1.97, 95% CI: 1.69, 2.29, p<0.001), Chattogram (APR: 3.59, 95% CI: 3.13, 4.12, p<0.001), Rajshahi (APR: 1.60, 95% CI: 1.37, 1.88, p<0.001), Barisal (APR: 2.83, 95% CI: 2.44, 3.29, p<0.001), Sylhet (APR: 4.50, 95% CI: 3.91, 5.17, p<0.001), Rangpur (APR: 1.95, 95% CI: 1.67, 2.28, p<0.001) and Mymensingh (APR: 1.40, 95% CI: 1.18, 1.66, p<0.001); living in non-slum urban area (APR: 1.75, 95% CI: 1.62, 1.89, p<0.001) and slum (APR: 1.74, 95% CI: 1.61, 1.88, p<0.001); belong to others religion category (APR: 1.10, 95% CI: 1.03, 1.17, p<0.01); being others (never married, separated, divorce) (APR: 1.11, 95% CI: 1.02, 1.21, p<0.05); having primary education (APR: 1.09, 95% CI: 1.01, 1.17, p<0.05), secondary education (APR: 1.28, 95% CI: 1.18, 1.38, p<0.001), higher secondary and above education (APR: 1.41, 95% CI: 1.28, 1.55, p<0.001); belong to middle wealth quintile (APR: 1.12, 95% CI: 1.02, 1.23, p<0.05), richer (APR: 1.17, 95% CI: 1.07, 1.29, p<0.01) and the richest quintile (APR: 1.18, 95% CI: 1.07, 1.30, p<0.01); inadequate fruits and vegetables intake (APR: 1.11, 95% CI: 1.00, 1.23, p<0.05); more than seven hours of sedentary time (APR: 1.09, 95% CI: 1.02, 1.17, p<0.05); being overweight/

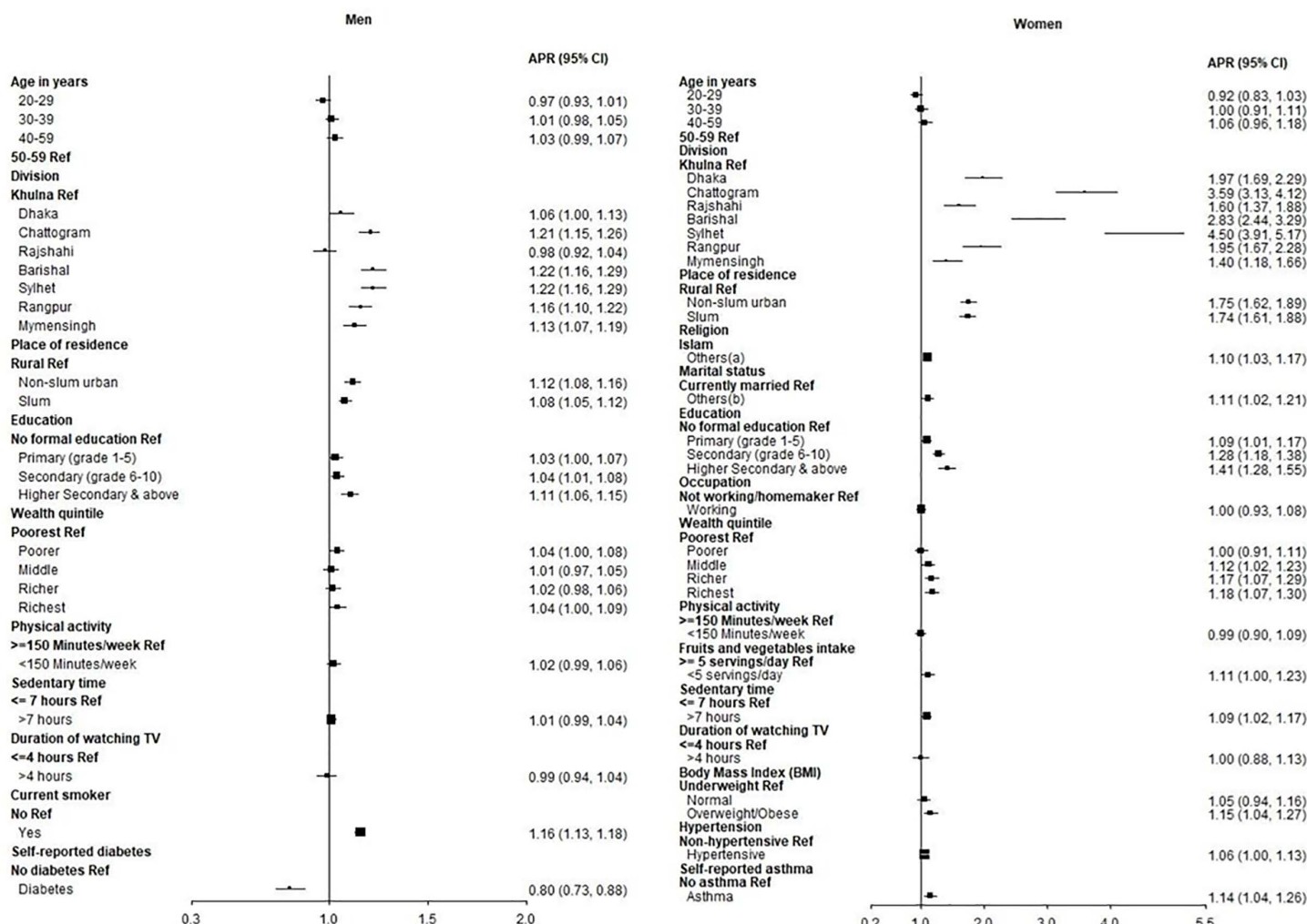

**Fig 5. Forest plot of adjusted prevalence ratios (APR) for factors of SSBs consumption among men and women.**

obese (APR: 1.15, 95% CI: 1.04, 1.27, p < 0.01); being hypertensive (APR: 1.06, 95% CI: 1.00, 1.13, p < 0.05); and self-reported asthma (APR: 1.14, 95% CI: 1.04, 1.26, p < 0.01) were associated with higher prevalence of SSBs consumption in the last week (Fig 5).

Bonferroni corrected p values and confidence interval (CI) for SSBs among adult men and women is provided in S10 Table.

## Discussion

In this cross-sectional study, the prevalence of different types of unhealthy food consumption was estimated and factors associated with unhealthy food consumption, such as SFS, SS, and SSBs, were identified. The findings indicated that a larger proportion of respondents consumed SFS, SS, and SSBs at least once a week and the prevalence of consumption was higher among men than among women. In addition, consumption of all three food types was more prevalent among the younger age group (20–29 years), with the exception of SSBs among men, participants with higher secondary and above education categories, in the highest wealth quintile group, and those who were overweight or obese. In our study, we also found that various sociodemographic, behavioral, and clinical factors were associated with the consumption of these food types. Education and wealth quintiles were associated with a higher consumption of SFS, SS, and SSBs for both men and women. Additionally, both men and women residing in non-slum urban and slum areas showed a higher association with the consumption of SFS and SSBs. Age (20–39 years) was associated with a higher SFS intake in both men and women. Moreover, current smoking was positively associated with unhealthy food consumption only among men. Overweight/obesity was found to be a contributing factor to the intake of SS and SSBs only in women. Furthermore, we found that a higher consumption of SSBs was associated with factors such as marital status, inadequate fruit and vegetable intake, overweight or obesity, high sedentary time, and self-reported asthma only in women.

### Prevalence of SFS, SS and SSBs consumption

In this study, a higher proportion of adult men consumed SFS and SS than women. A study conducted among elderly people from the same study areas in Bangladesh also reported that a higher proportion of elderly males consumed these foods compared to elderly females [33]. Corroborating comparisons with other studies are difficult because of the variations in food types, settings, timelines, and age groups across different studies. However, a study conducted in Bangladesh on ready-to-eat food consumption revealed that savory and fried snacks such as chips, *singara, somucha, peyaju*, and *beguni* were consumed by 8.1%, 6.8%, 2.1%, 44.3%, and 32.3% of adults, respectively [12]. In contrast, the proportion of adults who consumed sweets, such as cake, chocolate, large packet biscuits, and *jilapi,* was 14.5%, 2.1%, 22.6%, and 33.6%, respectively [12]. A study conducted in India reported that 50% of adults (aged 30 to ≥60 years) consumed savory foods 3–5 times a week [34]; however, this study did not provide any further segregation of the prevalence of savory food consumption between men and women.

In this study, more than three-quarters of adult men consumed SSB compared to only one-third of women. In another study conducted among elderly people from the same locations in Bangladesh, it was reported that a much higher proportion of elderly males consumed SSB (69%), while only 34% of elderly females consumed SSB [33]. In India, a recent study found that 96.3% of adults consumed SSBs [35]. Nevertheless, we cannot compare our findings with this study because it did not estimate the prevalence of SSB consumption by sex, and the assessment of SSB intake was based on daily, weekly, occasional, and never categories [35]. The findings from Burkina Faso and Kenya, two countries in sub-Saharan Africa, revealed that 50.38% of women consumed SSBs [36]. However, this prevalence is inconsistent with the observed prevalence of SSB intake among women in the current study. This discrepancy might be due to the variation in age range, as the present study focused on women aged 20–59 years, whereas Burkina Faso and Kenya included women aged 15–49 years [36]. Additionally, the operationalization of SSBs could be a contributing factor, as our study utilized 7-day dietary recalls, whereas Burkina Faso and Kenya employed 24-hour dietary recalls [36].

## Factors associated with SFS, SS and SSBs consumption

In our study, both men and women living in non-slum urban areas and slum areas had a higher prevalence of SFS and SSBs consumption than those living in rural areas. This finding is similar to that a of study conducted in Bangladesh (on ready-to-eat packaged food), India (on aerated drinks), and South Africa (on SSBs), where the consumption of these food types was more concentrated in urban areas [12,16,37]. One possible explanation could be that, in recent decades, Bangladesh's population has become more urban-centric, leading to a higher consumption of fast food due to busy schedules and lack of time to prepare meals at home [38]. Thus, urban areas should be considered during policy or program implementation in order to reduce this type of food consumption.

We found that educational level was a significant predictor of SFS, SS, and SSBs intake in both men and women. This is consistent with findings in and out of Bangladesh [16,38,39]. In Kansas City, education was found to be related to a higher intake of fast food among older adults [40]. Nevertheless, another study by Anderson et al. [41] pointed out that in the US, 18–64 years aged adults showed no association with education and consumption of fast food. However, a possible reason could be that in Bangladesh, with increasing education, females are more engaged in income-earning activities, leaving them with less time to prepare food, making it more convenient to consume unhealthy food. Wealth status was another factor associated with the consumption of SFS, SS, and SSBs regardless of sex. A recent study in Bangladesh among adolescents showed that a higher consumption of SFS was associated with the highest wealth quintile. Our result is akin to that of a study carried out in India, where they found the wealth index to be an associated factor for SSBs intake [16]. However, a study in Bangladesh reported that rich households spend less on food away from home [42]. In addition, a national longitudinal survey of adults on fast food consumption in the US revealed less consumption of fast food among adults in the highest quintile group [43]. However, the association found in our study might be that wealthy people have the affordability to buy and want to consume these food items more.

Our study revealed that current smokers, particularly men, were more likely to consume SFS, SS, and SSBs. There is evidence that smokers tend to have unhealthy diets and unhealthy patterns of food consumption [44,45]. Smoking may alter taste perception, leading individuals to add more salt and sugar to their food, as found in a study from the United Kingdom [46]. Studies conducted in India and Saudi Arabia have supported our findings on the association between current smoking and the consumption of SSBs [47]. Thus, integrating nutritional education into smoking cessation programs is necessary to promote healthy food consumption. In this study, overweight and/or obese women were more likely to consume SS and SSBs. A study conducted in Bangladesh among young adults and adults in the US revealed a significant association between fast-food consumption and obesity [41,48]. Our findings are similar to those of a study conducted in Korea, where it was reported that consumption of SSBs was associated with obesity among women [49]. Furthermore, a study conducted in Korea among women aged 18–81 years found that SSB consumption was related to obesity [50]. US findings from a cohort study reported that women with increased SSBs intake had increased weight and BMI [51].

We identified inadequate fruit and vegetable intake as being associated with SSB consumption among women, which supports the finding from another study that SSB consumption is high and there is less intake of fruits and vegetables [52]. Mathur et al. [16] mentioned that in LMICs among those who consumed SSBs or snacks the fruit and vegetable intake is lower among them. Thus, there is a need for nutrition education targeting women in Bangladesh. In the current study, SSBs consumption among women was related to increased sedentary time. A study conducted by Pengpid and Peltzer [37] among adults in South Africa found an association between sweetened fruit juice consumption and sedentary time. In the Asian context, studies on SSBs consumption were mostly conducted among children; however, an association was found between SSBs consumption and sedentary time among children [53]. We also found an association between asthma and SSBs consumption in women. A study showed that among adults, higher odds of having asthma was associated with SSB consumption ≥ 2 times/day [54]. Our findings showed an association between hypertension and SSBs consumption in women. This finding supports a recent cohort study conducted by Borresen et al. [55] among Norwegian women, where SSBs consumption was correlated with a higher risk of hypertension. In this study, we

observed an inverse association between diabetes and SSB consumption in men. A similar finding was observed among men in Saudi Arabia, where participants with diabetes consumed less SSBs than those who were non-diabetic [56]. Another study among US adults found that individuals with undiagnosed diabetes consumed more SSBs than those with diagnosed diabetes [57]. This indicates that education on less sugar consumption has an impact on the dietary habits of people with diabetes [57].

Compared to other studies, this study did not find any association between occupation, insufficient physical activity [39], and watching television [16,35] and consumption of SSBs. Moreover, this study did not find an association between self-reported heart disease and savory and fried snacks, sweets, and SSBs. A possible reason for this could be that the individuals might have underreported these diseases.

### Strengths and limitations

This study has notable strengths, such as a nationally representative sample that ensures generalizability. To our knowledge, this is the first study in Bangladesh to examine the prevalence and associated factors of SFS, SS, and SSBs in women and men aged 20–59 years. Nevertheless, this study has some limitations that should be considered when interpreting the results. The consumption of SFS, SS, and SSBs was self-reported; therefore, there is a chance of social desirability bias. The questionnaire was based on the participants' consumption over the seven days preceding data collection. Additionally, almost all covariates were self-reported, which may have introduced recall bias. The binary classification of unhealthy food consumption (weekly versus none) might not have fully captured habitual dietary behavior. Moreover, we did not collect information on portion sizes and the quantity of unhealthy food consumption; therefore, we could not measure calorie intake. Furthermore, due to administrative causes and financial limitations, data could not be collected from the seven rural clusters; hence, a reduction in sample size might affect the accurate estimation of the study. In addition, temporality could not be established in a cross-sectional study. Finally, some residual confounding may remain due to unmeasured lifestyle and cultural factors.

### Conclusion

Our analysis revealed a high prevalence of savory and fried snacks, sweet snacks, and SSBs consumption among 20–59 years aged women and men in Bangladesh. It also confirmed that consumption was more prevalent among men across various geographical, socio-demographic, and behavioral categories than among women. In addition, we identified the association of several modifiable factors, such as inadequate fruit and vegetable intake, high sedentary time, high BMI, and smoking, with the consumption of unhealthy food, which needs to be taken into consideration while designing and implementing interventions. The government of Bangladesh can utilize these data to design and execute actions aimed at reducing the consumption of unhealthy foods and promoting the consumption of healthy foods, while also considering other measures such as taxes on sugar-sweetened beverages, front-of-pack labeling, and advertising restrictions on unhealthy foods. Besides, in further research, objective measurement of the proportion of energy from unhealthy foods should be addressed, as, in the long run, our primary goal is to reduce overweight, obesity and diet related noncommunicable diseases in Bangladesh.

### Supporting information

**S1 Table. Prevalence of SFS, SS and SSBs consumption (last 7 days) by geographical and socio-demographic stratum of the study participants.**
(DOCX)

**S2 Table. Multicollinearity and goodness-of-fit statistics for SFS consumption among adult men and women.**
(DOCX)

**S3 Table. Multicollinearity and goodness-of-fit statistics for SS consumption among adult men and women.**
(DOCX)

**S4 Table. Multicollinearity and goodness-of-fit statistics for SSBs consumption among adult men and women.**
(DOCX)

**S5 Table. Crude prevalence ratios (CPR) and adjusted prevalence ratios (APR) of the factors of SFS consumption among men and women.**
(DOCX)

**S6 Table. Crude prevalence ratios (CPR) and adjusted prevalence ratios (APR) of the factors of SS consumption among men and women.**
(DOCX)

**S7 Table. Crude prevalence ratios (CPR) and adjusted prevalence ratios (APR) of the factors of SSBs consumption among men and women.**
(DOCX)

**S8 Table. Crude prevalence ratios (CPR) and adjusted prevalence ratios (APR) of the factors of SFS consumption among men and women (Bonferroni corrected).**
(DOCX)

**S9 Table. Crude prevalence ratios (CPR) and adjusted prevalence ratios (APR) of the factors of SS consumption among men and women (Bonferroni corrected).**
(DOCX)

**S10 Table. Crude prevalence ratios (CPR) and adjusted prevalence ratios (APR) of the factors of SSBs consumption among men and women (Bonferroni corrected).**
(DOCX)

**S1 Data. Dataset in CSV format (https://doi.org/10.17605/OSF.IO/YJWNG).**
(CSV)

**S1 File. Questionnaire in English language.**
(PDF)

**S2 File. Questionnaire in Bengali language.**
(PDF)

## Acknowledgments

We express our gratitude to all the study participants, data collectors, field supervisors and managers involved in the national nutrition surveillance conducted during 2018–2019.

## Author contributions

**Conceptualization:** Shahnaz Sharmin, Fahmida Akter, Abu Ahmed Shamim, Malay Kanti Mridha.

**Data curation:** Md. Mokbul Hossain.

**Formal analysis:** Shahnaz Sharmin, Md. Mokbul Hossain.

**Funding acquisition:** Malay Kanti Mridha.

**Investigation:** Md. Mokbul Hossain, Abu Ahmed Shamim, Mehedi Hasan, Md Showkat Ali Khan, Mohammad Aman Ullah, Dipak Kumar Mitra, Malay Kanti Mridha.

**Methodology:** Md. Mokbul Hossain, Abu Ahmed Shamim, Mehedi Hasan, Md Showkat Ali Khan, Mohammad Aman Ullah, Dipak Kumar Mitra, Malay Kanti Mridha.

**Project administration:** Abu Abdullah Mohammad Hanif, Malay Kanti Mridha.

**Supervision:** Fahmida Akter, Md. Mokbul Hossain, Abu Ahmed Shamim, Abu Abdullah Mohammad Hanif, Malay Kanti Mridha.

**Writing – original draft:** Shahnaz Sharmin.

**Writing – review & editing:** Shahnaz Sharmin, Fahmida Akter, Md. Mokbul Hossain, Abu Ahmed Shamim, Abu Abdullah Mohammad Hanif, Mehedi Hasan, Md Showkat Ali Khan, Mohammad Aman Ullah, Dipak Kumar Mitra, Malay Kanti Mridha.

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
