## [Decision Letter · Decision Letter 0]

2 Jun 2025

Dear Dr. Rahman,

Thank you for submitting your manuscript to PLOS ONE. After careful consideration, we feel that it has merit but does not fully meet PLOS ONE’s publication criteria as it currently stands. Therefore, we invite you to submit a revised version of the manuscript that addresses the points raised during the review process.

We look forward to receiving your revised manuscript.

Kind regards,

Jordan Llego, PhD ELM, D. Hon. Ex., PhDN, RN

Academic Editor

PLOS ONE

Journal Requirements:

3. Please amend the manuscript submission data (via Edit Submission) to include author Malay Kanti Mridha.

4. Please amend your authorship list in your manuscript file to include author Tareq Rahman.

5. We are unable to open your Supporting Information file S1_Data.dta. Please kindly revise as necessary and re-upload.

Additional Editor Comments:

Thank you for submitting your manuscript to PLOS ONE. We appreciate your contribution to addressing the pressing issue of unhealthy food consumption in Bangladesh using nationally representative data. After a careful evaluation by a peer reviewer and an editorial assessment, we have decided on your submission. Your manuscript addresses an important and timely topic in public health, offering valuable insights into dietary behaviors and their associated sociodemographic and clinical factors within the context of low- and middle-income countries. However, substantial revisions are required to enhance the scientific rigor, clarity, and interpretability of your study, and we are recommending a major revision at this stage.

Several key concerns and required revisions need to be addressed. First, the binary classification of food consumption as weekly versus none lacks sensitivity in capturing habitual dietary behavior. We encourage you to justify this classification using supporting literature and consider including more granular frequency categories if available or acknowledge this limitation. Additionally, while your manuscript mentions model fitness, it is missing critical information. Please report variance inflation factors (VIFs) to assess multicollinearity and include goodness-of-fit statistics, such as the Pearson Chi-square or deviance, for your Poisson regression models.

There is also the issue of multiple comparisons adjustment; given the number of predictors examined, failing to adjust for multiple testing increases the risk of Type I error. Consider applying a correction method like Bonferroni or FDR or to justify its omission explicitly. Moreover, the manuscript contains inconsistent terminology (e.g., "junk food" versus "unhealthy food") as well as various grammatical and structural issues. We strongly recommend professional English language editing to enhance readability and coherence.

The Discussion section could be further developed, as it restates results without adequate interpretation. We encourage you to situate your findings within established behavioral or theoretical frameworks, offer gender-sensitive interpretations, and provide clear policy implications and recommendations for targeted interventions. Additionally, the current limitations section is insufficient; you should discuss self-report bias and recall issues, as well as cross-sectional design limitations and the lack of data on portion size and caloric intake.

Lastly, while your data availability statement claims full access, we encourage you to deposit the dataset in a public repository with a Digital Object Identifier (DOI) to promote transparency and reproducibility. The visual presentation of Tables 3 and 4 is also an area of concern; they are dense and would benefit from simplification. Consider adding figures such as bar charts or forest plots to enhance visual clarity.

We believe that addressing these revisions will strengthen your manuscript and make it suitable for publication in PLOS ONE. Please revise the manuscript accordingly and submit a detailed point-by-point response to each comment made by me and the reviewer. We look forward to receiving your revised submission.

Reviewers' comments:

Reviewer's Responses to Questions

**Comments to the Author**

1. Is the manuscript technically sound, and do the data support the conclusions?

Reviewer #1: Partly

2. Has the statistical analysis been performed appropriately and rigorously?

Reviewer #1: I Don't Know

3. Have the authors made all data underlying the findings in their manuscript fully available?

Reviewer #1: Yes

4. Is the manuscript presented in an intelligible fashion and written in standard English?

Reviewer #1: No

Reviewer #1: Thank you for the opportunity to review this important and timely manuscript titled “Unhealthy food consumption among 20–59-year-old adults in Bangladesh: findings from a nationally representative cross-sectional survey.” The study addresses a critical public health issue using nationally representative data and provides helpful information regarding sociodemographic, behavioral, and clinical determinants of unhealthy food consumption. Major revisions are necessary to strengthen the scientific rigor, clarity, and overall contribution of this manuscript. Below are detailed comments organized according to the evaluation criteria:

The simple yes-or-no classification of "weekly consumption" makes dietary behavior too basic and should either be explained better or changed to a more detailed measure that shows how often or how much is eaten.

Even though the manuscript talks about model diagnostics (page 8, lines 150–153), it does not provide variance inflation factors or goodness-of-fit statistics, which are needed to show that the model is good and to check for multicollinearity.

The analysis includes many comparisons without correcting for multiple tests (page 9, table 4), which raises the chance of getting false positives; think about using a correction method like Bonferroni or FDR.

The data availability statement confirms full access (page 2, line 35), but the authors are encouraged to enhance transparency by depositing the dataset in a public repository with a DOI.

The manuscript contains grammatical errors, awkward sentence structures, and inconsistent terminology—such as "junk food" versus "unhealthy food"—throughout, including on Page 3, Lines 40–44, and would benefit from professional English editing.

The discussion section (pages 17–18) primarily summarizes findings but lacks deeper interpretation through a behavioral or policy lens, particularly regarding gender differences and public health implications.

The limitations section (page 8, lines 154–160) is minimal; it should be expanded to include self-reporting bias, the cross-sectional nature of the study, and lack of data on portion sizes or frequency of consumption.

Tables, especially Table 3 (pages 13–15), are data-heavy; visual representations such as forest plots or bar charts should be added to improve clarity and reader comprehension.

**Do you want your identity to be public for this peer review?** For information about this choice, including consent withdrawal, please see our Privacy Policy

Reviewer #1: No

---

## [Author Response · Author response to Decision Letter 1]

21 Aug 2025

Additional Editor Comments

1. Thank you for submitting your manuscript to PLOS ONE. We appreciate your contribution to addressing the pressing issue of unhealthy food consumption in Bangladesh using nationally representative data. After a careful evaluation by a peer reviewer and an editorial assessment, we have decided on your submission. Your manuscript addresses an important and timely topic in public health, offering valuable insights into dietary behaviors and their associated sociodemographic and clinical factors within the context of low- and middle-income countries. However, substantial revisions are required to enhance the scientific rigor, clarity, and interpretability of your study, and we are recommending a major revision at this stage.

Response: Thank you for your suggestion. We have revised the manuscript as per your recommendations. Please find the details below.

2. Several key concerns and required revisions need to be addressed. First, the binary classification of food consumption as weekly versus none lacks sensitivity in capturing habitual dietary behavior. We encourage you to justify this classification using supporting literature and consider including more granular frequency categories if available or acknowledge this limitation. Additionally, while your manuscript mentions model fitness, it is missing critical information. Please report variance inflation factors (VIFs) to assess multicollinearity and include goodness-of-fit statistics, such as the Pearson Chi-square or deviance, for your Poisson regression models.

Response: Thank you for your observation. We agree that categorizing food consumption simply as weekly versus none does not adequately reflect habitual dietary behavioral patterns. However, there are several papers that have used binary classification of food consumption for analysis (https://pubmed.ncbi.nlm.nih.gov/35392287/, https://pubmed.ncbi.nlm.nih.gov/32029986/, https://pubmed.ncbi.nlm.nih.gov/12833112/). Based on these studies, we developed this binary classification.

As per your suggestion, we have reported multicollinearity and goodness-of-fit statistics for all models considered for three food groups (savory and fried snacks, SFS; sweet snacks, SS; and sugar-sweetened beverages, SSBs), which can be found in supporting Tables S2, S3, and S4. Result section of the manuscript is also updated accordingly (page 17, lines 219-226). The text stated as follows:

Multicollinearity and goodness-of-fit statistics: Multicollinearity statistics for explanatory variables and goodness-of-fit statistics for SFS, SS, and SSBs, separately for adult men and women, are presented in Tables S2, S3, and S4, respectively. The mean VIF values are all less than 10 for all models considered, indicating no multicollinearity among the explanatory variables. Both deviance and Pearson Chi-square statistics showed large p values, indicating there is no lack of fit and suggesting that the Poisson regression models fit the data well.

3. There is also the issue of multiple comparisons adjustment; given the number of predictors examined, failing to adjust for multiple testing increases the risk of Type I error. Consider applying a correction method like Bonferroni or FDR or to justify its omission explicitly. Moreover, the manuscript contains inconsistent terminology (e.g., "junk food" versus "unhealthy food") as well as various grammatical and structural issues. We strongly recommend professional English language editing to enhance readability and coherence.

Response: Thank you for your valuable comment. We used 18 predictors in the model and considered p-values <0.05 as statistically significant. Given this, there is less than a 5% chance that one predictor would be significant by chance. Moreover, for many of our factors, the p-value was <0.001. For this reason, we did not apply Bonferroni or FDR-based corrections.

In this manuscript, we used the term ‘unhealthy food’ to represent (savory and fried snacks, SFS; sweet snacks, SS; and sugar-sweetened beverages, SSBs). We used other terms, such as junk food and fast food, only in the introduction and discussion sections to introduce and explain the broader range of unhealthy foods mentioned by other authors and sources (https://link.springer.com/article/10.1007/s10597-020-00672-x, https://pubmed.ncbi.nlm.nih.gov/38482184/, https://pubmed.ncbi.nlm.nih.gov/35816403/). Regarding English editing we checked and edited the manuscript to correct grammatical errors.

4. The Discussion section could be further developed, as it restates results without adequate interpretation. We encourage you to situate your findings within established behavioral or theoretical frameworks, offer gender-sensitive interpretations, and provide clear policy implications and recommendations for targeted interventions. Additionally, the current limitations section is insufficient; you should discuss self-report bias and recall issues, as well as cross-sectional design limitations and the lack of data on portion size and caloric intake.

Response: We revised and enriched the whole discussion section as per your recommendation. We have added new text also (highlighted in yellow).

Thank you for your observation. We elaborated the section as you suggested. We added social desirability bias (please see line 443, page 34). We mentioned recall bias earlier (please see line 445, page 34), the cross-sectional nature of the study (line 452, page 34), and the calorie intake issue (line 449, page 34).

5. Lastly, while your data availability statement claims full access, we encourage you to deposit the dataset in a public repository with a Digital Object Identifier (DOI) to promote transparency and reproducibility. The visual presentation of Tables 3 and 4 is also an area of concern; they are dense and would benefit from simplification. Consider adding figures such as bar charts or forest plots to enhance visual clarity.

Response: Thank you for your comment. Depositing the dataset in Dryad requires a data publishing fee, which we are unable to provide. However, we have shared the dataset as supporting information in both CSV and STATA formats.

As recommended, we have replaced Table 3 with a bar chart (now Fig 2) to present prevalence by gender and area of residence and moved the full table to the supplementary table (S1 Table). However, we would prefer to retain Table 4 in the main manuscript, as it presents both crude and adjusted prevalence ratios, which are central to our analysis and important for readers to interpret the key findings in context.

Reviewer#1 comment

1. Thank you for the opportunity to review this important and timely manuscript titled “Unhealthy food consumption among 20–59-year-old adults in Bangladesh: findings from a nationally representative cross-sectional survey.” The study addresses a critical public health issue using nationally representative data and provides helpful information regarding sociodemographic, behavioral, and clinical determinants of unhealthy food consumption. Major revisions are necessary to strengthen the scientific rigor, clarity, and overall contribution of this manuscript. Below are detailed comments organized according to the evaluation criteria:

Response: Thank you for your comment. We have revised the manuscript based on your suggestions. Please find the details below.

2. The simple yes-or-no classification of "weekly consumption" makes dietary behavior too basic and should either be explained better or changed to a more detailed measure that shows how often or how much is eaten.

Response: Thank you for your valuable comment. We agree that categorizing food consumption simply as weekly versus none does not adequately reflect habitual dietary behavioral patterns. However, there are several papers that have used binary classification of food consumption for analysis (https://pubmed.ncbi.nlm.nih.gov/35392287/, https://pubmed.ncbi.nlm.nih.gov/32029986/, https://pubmed.ncbi.nlm.nih.gov/12833112/). Based on these studies, we developed this binary classification.

3. Even though the manuscript talks about model diagnostics (page 8, lines 150–153), it does not provide variance inflation factors or goodness-of-fit statistics, which are needed to show that the model is good and to check for multicollinearity.

Response: Thank you for your comment. As per your suggestion, we have reported multicollinearity and goodness-of-fit statistics for all models considered for three food groups (savory and fried snacks, SFS; sweet snacks, SS; and sugar-sweetened beverages, SSBs), which can be found in supporting Tables S2, S3, and S4. Result section of the manuscript is also updated accordingly (page 17, lines 219-226). The text stated as follows:

Multicollinearity and goodness-of-fit statistics

Multicollinearity statistics for explanatory variables and goodness-of-fit statistics for SFS, SS, and SSBs, separately for adult men and women, are presented in Tables S2, S3, and S4, respectively. The mean VIF values are all less than 10 for all models considered, indicating no multicollinearity among the explanatory variables. Both deviance and Pearson Chi-square statistics showed large p values, indicating there is no lack of fit and suggesting that the Poisson regression models fit the data well.

4. The analysis includes many comparisons without correcting for multiple tests (page 9, table 4), which raises the chance of getting false positives; think about using a correction method like Bonferroni or FDR.

Response: Thank you for your observation. We used 18 predictors in the model and considered p-values <0.05 as statistically significant. Given this, there is less than a 5% chance that one predictor would be significant by chance. Moreover, for many of our factors, the p-value was <0.001. For this reason, we did not apply Bonferroni or FDR-based corrections.

5. The data availability statement confirms full access (page 2, line 35), but the authors are encouraged to enhance transparency by depositing the dataset in a public repository with a DOI.

Response: Thank you for your comment. Depositing the dataset in Dryad requires a data publishing fee, which we are unable to provide. However, we have shared the dataset as supporting information in both CSV and STATA formats.

6. The manuscript contains grammatical errors, awkward sentence structures, and inconsistent terminology—such as "junk food" versus "unhealthy food"—throughout, including on Page 3, Lines 40–44, and would benefit from professional English editing.

Response: Thank you for your observation. However, in this manuscript, we used the term ‘unhealthy food’ to represent (savory and fried snacks, SFS; sweet snacks, SS; and sugar-sweetened beverages, SSBs). We used other terms, such as junk food and fast food, only in the introduction and discussion sections to introduce and explain the broader range of unhealthy foods mentioned by other authors and sources (https://link.springer.com/article/10.1007/s10597-020-00672-x, https://pubmed.ncbi.nlm.nih.gov/38482184/, https://pubmed.ncbi.nlm.nih.gov/35816403/). Regarding English editing we checked and edited the manuscript to correct grammatical errors.

7. The discussion section (pages 17–18) primarily summarizes findings but lacks deeper interpretation through a behavioral or policy lens, particularly regarding gender differences and public health implications.

Response: We revised the whole discussion section as per your recommendation. We have added new text also (highlighted in yellow).

8. The limitations section (page 8, lines 154–160) is minimal; it should be expanded to include self-reporting bias, the cross-sectional nature of the study, and lack of data on portion sizes or frequency of consumption.

Response: Thank you for your comments on limitation section. We elaborated on the limitations section and included points on social desirability bias (line 443, page 34), recall bias (line 445, page 34), the cross-sectional nature of the study (line 452, page 34), and the lack of data on portion sizes and calorie intake (line 449, page 34).

9. Tables, especially Table 3 (pages 13–15), are data-heavy; visual representations such as forest plots or bar charts should be added to improve clarity and reader comprehension.

Response: Thank you for your comments. As recommended, we have replaced Table 3 with a bar chart (now Fig 2) to present prevalence by gender and area of residence and moved the full table to the supplementary section (S1 Table).

---

## [Decision Letter · Decision Letter 1]

11 Sep 2025

Dear Dr. Mridha,

Thank you for submitting your manuscript to PLOS ONE. After careful consideration, we feel that it has merit but does not fully meet PLOS ONE’s publication criteria as it currently stands. Therefore, we invite you to submit a revised version of the manuscript that addresses the points raised during the review process.

We look forward to receiving your revised manuscript.

Kind regards,

Jordan Llego, PhD ELM, D. Hon. Ex., PhDN, RN

Academic Editor

PLOS ONE

Journal Requirements:

Additional Editor Comments :

Thank you for submitting your revised manuscript to PLOS ONE. We appreciate the thoughtful and thorough responses you've provided to both reviewer and editorial comments. The improvements you’ve made—such as adding diagnostic statistics, refining the discussion, and including visual presentations for prevalence estimates—are clear and valuable. Your study tackles an important public health issue using nationally representative data, which makes the findings relevant to a wide audience. That said, there are still a few areas that need further revision before your manuscript can be considered for publication.

Although you have included CSV and STATA files as supplementary material, we strongly recommend depositing your dataset in a recognized open-access public repository, such as Zenodo or OSF, and obtaining a DOI. This step will promote transparency, increase accessibility, and align your work with best practices in open science. In the methods section, the criteria for defining sufficient versus insufficient physical activity are unclear—the thresholds seem inverted and do not match the WHO guidelines. Please clarify the exact criteria you used and, if they differ from WHO recommendations, provide a clear justification for your approach.

You've included variance inflation factors and model fit statistics in the supplementary tables, which is helpful, but summarizing these results briefly in the main Results section would make the findings more accessible. Including mean VIF values and p-values for deviance and Pearson chi-square directly in the narrative will help readers better understand the robustness of your models. Additionally, the current explanation for not adjusting for multiple comparisons is not sufficient. With so many predictors, the risk of Type I error is real. We suggest either applying a statistical correction, such as Bonferroni or FDR, or providing a more compelling justification for not using these adjustments.

There are still a few minor grammatical errors, such as “foods intake” instead of “food intake,” and the terminology could be more consistent—consider using “unhealthy food” throughout. A final professional language edit is recommended to help the manuscript read smoothly. While the revised discussion is stronger, it would be even more effective if it were more clearly grounded in behavioral or theoretical frameworks, like nutrition transition or health belief models. Expanding on gender-sensitive interpretations of your results would also add depth. In the policy implications section, try to move beyond general recommendations and suggest specific, practical strategies for Bangladesh, such as taxing sugar-sweetened beverages, introducing front-of-pack labeling, or implementing advertising restrictions. The limitations section is better, but could be further improved by openly acknowledging the likelihood of residual confounding due to unmeasured lifestyle, cultural, or marketing factors.

On presentation, the switch from Table 3 to a bar chart is a great improvement and adds clarity. However, Table 4 is still dense and difficult to interpret. Consider breaking it into smaller, more focused tables, or adding forest plots to make the information clearer for readers. Finally, the funding statement mentions that funders were involved in study design and review. Please clarify exactly how the funders participated and discuss whether their involvement could have introduced bias, so readers have full transparency about this aspect of the research.

Reviewers' comments:

Reviewer's Responses to Questions

**Comments to the Author**

Reviewer #1: All comments have been addressed

2. Is the manuscript technically sound, and do the data support the conclusions?

Reviewer #1: Yes

3. Has the statistical analysis been performed appropriately and rigorously?

Reviewer #1: Yes

4. Have the authors made all data underlying the findings in their manuscript fully available?

Reviewer #1: Yes

5. Is the manuscript presented in an intelligible fashion and written in standard English?

Reviewer #1: No

Reviewer #1: Thank you for submitting your manuscript. Below is my comment:

1. In data availability, please consider depositing the dataset (and code if possible) in a free public repository (e.g., Zenodo, OSF) to obtain a DOI, which will enhance transparency and accessibility.

2. Methods – Physical Activity Definition, kindly clarify the definition and thresholds for sufficient physical activity to ensure alignment with WHO guidelines, or justify the criteria used.

3. Statistical Reporting. Kindly summarize key diagnostic results (mean VIF ranges, p-values for model fit) in the main text for clarity, while retaining full details in the supplementary tables.

4. Language and Terminology, maybe conduct a final proofreading pass to correct minor grammatical issues (e.g., “foods intake” to “food intake”) and ensure consistent use of the term “unhealthy food.”

5. Discussion and Limitations, kindly add a brief note on potential residual confounding (e.g., unmeasured lifestyle or marketing influences) and acknowledge the bias direction from using binary food classifications.

**Do you want your identity to be public for this peer review?** For information about this choice, including consent withdrawal, please see our Privacy Policy

Reviewer #1: No

---

## [Author Response · Author response to Decision Letter 2]

27 Oct 2025

Manuscript ID: PONE-D-25-00015

Manuscript title: Unhealthy food consumption among 20-59 years old adults in Bangladesh: findings from a nationally representative cross-sectional survey

Additional Editor Comments

1. Thank you for submitting your revised manuscript to PLOS ONE. We appreciate the thoughtful and thorough responses you've provided to both reviewer and editorial comments. The improvements you’ve made—such as adding diagnostic statistics, refining the discussion, and including visual presentations for prevalence estimates—are clear and valuable. Your study tackles an important public health issue using nationally representative data, which makes the findings relevant to a wide audience. That said, there are still a few areas that need further revision before your manuscript can be considered for publication.

Response: Thank you for your suggestion. We have revised the manuscript as per your recommendations. Please find the details below.

2. Although you have included CSV and STATA files as supplementary material, we strongly recommend depositing your dataset in a recognized open-access public repository, such as Zenodo or OSF, and obtaining a DOI. This step will promote transparency, increase accessibility, and align your work with best practices in open science. In the methods section, the criteria for defining sufficient versus insufficient physical activity are unclear—the thresholds seem inverted and do not match the WHO guidelines. Please clarify the exact criteria you used and, if they differ from WHO recommendations, provide a clear justification for your approach.

Response: As per your suggestion, we have deposited the dataset to OSF, and the DOI is https://doi.org/10.17605/OSF.IO/YJWNG

Thank you for your observation. We apologize for this inversion, which occurred due to a typing error in one place in the methods section (Table 1), where instead of “≥150 minutes,” we mistakenly wrote “<150 minutes” as sufficient, while it should have been insufficient. We defined physical activity according to the WHO guideline as follows: Insufficient (<150 minutes of moderate-intensity or <75 minutes of vigorous-intensity physical activity or an equivalent combination of both in a week) and Sufficient (≥150 minutes of moderate-intensity or ≥75 minutes of vigorous-intensity physical activity or an equivalent combination of both in a week) (https://iris.who.int/server/api/core/bitstreams/d0972fd5-8f7d-4c87-b092-889e0f5f4618/content).

This threshold has also been used in other studies, including (https://journals.plos.org/plosone/article?id=10.1371/journal.pone.0251967). We have corrected this typo in Table 1 (Page 7, line 141) of the methods section.

3. You've included variance inflation factors and model fit statistics in the supplementary tables, which is helpful, but summarizing these results briefly in the main Results section would make the findings more accessible. Including mean VIF values and p-values for deviance and Pearson chi-square directly in the narrative will help readers better understand the robustness of your models. Additionally, the current explanation for not adjusting for multiple comparisons is not sufficient. With so many predictors, the risk of Type I error is real. We suggest either applying a statistical correction, such as Bonferroni or FDR, or providing a more compelling justification for not using these adjustments.

Response: We have summarized the mean VIF and the p-values for deviance and Pearson chi-square in the result section (Page 13-14, lines 216–222).

Thank you for your comment. We want to mention that our study is an exploratory study. For an exploratory study, it is not necessary to perform a Bonferroni adjustment (https://pubmed.ncbi.nlm.nih.gov/11297884/). Moreover, six predictors with more than two categories are considered in the regression model (age, division, education, wealth quintile, place of residence, and BMI). We compared each category with the reference category and did not make any conclusions based on multiple pairwise tests or confidence intervals.

However, we still obtained Bonferroni-adjusted confidence intervals and p-values, which are provided in the supplementary table (S8, S9, and S10 Table). We did not observe any substantial changes (except for the wealth quintile comparisons between the poorest and richer, and poorest and richest for men; the poorest and poorer, and poorest and middle for women in savory and fried snacks (SFS); the poorest and middle for men, and no formal education compared with primary for women in sweet snacks (SS); and no formal education compared with secondary, poorest compared with highest wealth quintile for men, and poorest compared with middle for women in sugar-sweetened beverages (SSBs)) between the Bonferroni-adjusted p-values and those in the main regression tables; hence, we retained the main table.

4. There are still a few minor grammatical errors, such as “foods intake” instead of “food intake,” and the terminology could be more consistent—consider using “unhealthy food” throughout. A final professional language edit is recommended to help the manuscript read smoothly. While the revised discussion is stronger, it would be even more effective if it were more clearly grounded in behavioral or theoretical frameworks, like nutrition transition or health belief models. Expanding on gender-sensitive interpretations of your results would also add depth. In the policy implications section, try to move beyond general recommendations and suggest specific, practical strategies for Bangladesh, such as taxing sugar-sweetened beverages, introducing front-of-pack labeling, or implementing advertising restrictions. The limitations section is better, but could be further improved by openly acknowledging the likelihood of residual confounding due to unmeasured lifestyle, cultural, or marketing factors.

Response: Thank you for your suggestion. We have checked and edited the manuscript to correct grammatical errors. We also revised and enriched the discussion section as per your recommendation and have added new text (highlighted in yellow). We mentioned residual confounding in the limitations section (Page 23, line 432).

5. On presentation, the switch from Table 3 to a bar chart is a great improvement and adds clarity. However, Table 4 is still dense and difficult to interpret. Consider breaking it into smaller, more focused tables, or adding forest plots to make the information clearer for readers. Finally, the funding statement mentions that funders were involved in study design and review. Please clarify exactly how the funders participated and discuss whether their involvement could have introduced bias, so readers have full transparency about this aspect of the research.

Response: Thank you for your comment. As you suggested we have replaced table 3,4 and 5 as forest plot (now Fig 3, Fig 4, and Fig 5). However, we moved these tables to the supplementary table (S5 Table, S6 Table, and S7 Table) so that readers can also see the crude prevalence ratio.

Regarding the funders’ involvement, we would like to clarify that they suggested some additional questions for the main questionnaire, attended meetings on data supervision and project updates, and reviewed the manuscript. Their involvement did not affect the study outcomes or interpretation, as all final decisions about the study design, data analysis, and manuscript content were made by the investigators. This ensures transparency and maintains the integrity of the research.

Reviewer#1 comment

1. In data availability, please consider depositing the dataset (and code if possible) in a free public repository (e.g., Zenodo, OSF) to obtain a DOI, which will enhance transparency and accessibility.

Response: Thank you for your suggestion. We have deposited the dataset to OSF, and the DOI is https://doi.org/10.17605/OSF.IO/YJWNG

2. Methods – Physical Activity Definition, kindly clarify the definition and thresholds for sufficient physical activity to ensure alignment with WHO guidelines, or justify the criteria used.

Response: We defined physical activity according to the WHO guideline as follows: insufficient (<150 minutes of moderate-intensity or <75 minutes of vigorous-intensity physical activity or an equivalent combination of both in a week) and sufficient (≥150 minutes of moderate-intensity or ≥75 minutes of vigorous-intensity physical activity or an equivalent combination of both in a week) (https://iris.who.int/server/api/core/bitstreams/d0972fd5-8f7d-4c87-b092-889e0f5f4618/content).

This threshold has also been used in other studies, including (https://journals.plos.org/plosone/article?id=10.1371/journal.pone.0251967). However, due to a typing error in one place in the methods section (Table 1), instead of “≥150 minutes,” we mistakenly wrote “<150 minutes” as sufficient, while it should have been insufficient. We have corrected this typo in Table 1 (Page 7, line 141) of the methods section.

3. Statistical Reporting. Kindly summarize key diagnostic results (mean VIF ranges, p-values for model fit) in the main text for clarity, while retaining full details in the supplementary tables.

Response: We have summarized the mean VIF and the p-values for deviance and Pearson chi-square in the result section (Page 13-14, lines 216–222).

4. Language and Terminology, maybe conduct a final proofreading pass to correct minor grammatical issues (e.g., “foods intake” to “food intake”) and ensure consistent use of the term “unhealthy food.”

Response: Thank you for your comment. We have checked and edited the manuscript to correct grammatical errors. In this manuscript, we used the term ‘unhealthy food’ to represent (savory and fried snacks, SFS; sweet snacks, SS; and sugar-sweetened beverages, SSBs). We used other terms, such as junk food and fast food, only in the introduction and discussion sections to introduce and explain the broader range of unhealthy foods mentioned by other authors and sources (https://link.springer.com/article/10.1007/s10597-020-00672-x, https://pubmed.ncbi.nlm.nih.gov/38482184/, https://pubmed.ncbi.nlm.nih.gov/35816403/).

5. Discussion and Limitations, kindly add a brief note on potential residual confounding (e.g., unmeasured lifestyle or marketing influences) and acknowledge the bias direction from using binary food classifications.

Response: Thank you for your suggestion. We added residual confounding (please see page 23, line 431) and binary food classification (page 23, line 426).

---

## [Decision Letter · Decision Letter 2]

4 Nov 2025

Unhealthy food consumption among 20-59 years old adults in Bangladesh: findings from a nationally representative cross-sectional survey

PONE-D-25-00015R2

Dear Dr. Mridha,

We’re pleased to inform you that your manuscript has been judged scientifically suitable for publication and will be formally accepted for publication once it meets all outstanding technical requirements.

Kind regards,

Jordan Llego, PhD ELM, D. Hon. Ex., PhDN, RN

Academic Editor

PLOS ONE

Additional Editor Comments (optional):

Thank you for sending the updated version of your manuscript to PLOS ONE. I’m happy to let you know that, after carefully reviewing your changes and your thoughtful responses to the reviewers’ and editors’ feedback, your paper has been accepted for publication as it is.

Your team has made impressive improvements since the first submission. The way you clarified the methodology—especially updating the definition of physical activity to follow WHO guidelines and adding variance inflation factors and model fit diagnostics—shows great attention to detail and transparency. Including the OSF dataset with a DOI also makes your research more reproducible and open, which is exactly what PLOS ONE’s data-sharing policy encourages.

Now, your manuscript reads clearly and flows well, with more consistent language and technical accuracy. The improved discussion section weaves in important theories—like the nutrition transition and health belief models—which help make sense of the gender-specific findings. Adding forest plots to show your results also makes them easier to understand and more engaging for readers.

You also took care to address the editorial feedback—fixing small grammatical issues, expanding on policy implications, and explaining the funders’ role—which has made your manuscript both polished and transparent. Your statements about ethics, data availability, and competing interests all meet the journal’s requirements.

Given all these improvements and the strong scientific contribution of your nationally representative analysis, I’m happy to tell you that your manuscript is accepted for publication with no further changes needed.

Congratulations on this accomplishment, and thank you for choosing PLOS ONE to share your important findings about unhealthy food consumption and its causes in Bangladesh. We look forward to sharing your work with the worldwide public health and nutrition research community.

Reviewers' comments:

Reviewer's Responses to Questions

**Comments to the Author**

Reviewer #1: All comments have been addressed

2. Is the manuscript technically sound, and do the data support the conclusions?

Reviewer #1: Yes

3. Has the statistical analysis been performed appropriately and rigorously?

Reviewer #1: Yes

4. Have the authors made all data underlying the findings in their manuscript fully available?

Reviewer #1: Yes

5. Is the manuscript presented in an intelligible fashion and written in standard English?

Reviewer #1: Yes

Reviewer #1: Thank you for submitting your manuscript to this journal. You addressed prior concerns. The design, sampling, and ethics are explicit. Psychometric reporting is adequate for a screening index. Key diagnostics are in the text and tables. The scoring approach and interpretation are clear. Claims are aligned to the data and the adolescent sample. Figures and tables read well. References follow journal style. No concerns about ethics or duplicate publication.

**Do you want your identity to be public for this peer review?** For information about this choice, including consent withdrawal, please see our Privacy Policy

Reviewer #1: No

---

## [Editor Report · Acceptance letter]

PONE-D-25-00015R2

PLOS ONE

Dear Dr. Mridha,

I'm pleased to inform you that your manuscript has been deemed suitable for publication in PLOS ONE. Congratulations! Your manuscript is now being handed over to our production team.

Kind regards,

on behalf of

Dr. Jordan Llego

Academic Editor

PLOS ONE